# A Method for the Precise Coordinate Determination of an Inaccessible Location

**DOI:** 10.3390/s23198199

**Published:** 2023-09-30

**Authors:** Edward Osada, Magdalena Owczarek-Wesołowska, Krzysztof Karsznia, Kazimierz Becek, Zbigniew Muszyński

**Affiliations:** 1Faculty of Applied Studies, University of Lower Silesia, 53-611 Wroclaw, Poland; edward.osada@dsw.edu.pl; 2Faculty of Environmental Engineering and Geodesy, Wroclaw University of Environmental and Life Sciences, 50-355 Wroclaw, Poland; magdalena.owczarek-wesolowska@upwr.edu.pl; 3Faculty of Geodesy and Cartography, Warsaw University of Technology, 00-661 Warsaw, Poland; krzysztof.karsznia@pw.edu.pl; 4Faculty of Geoengineering, Mining and Geology, Wroclaw University of Science and Technology, 50-421 Wroclaw, Poland; zbigniew.muszynski@pwr.edu.pl

**Keywords:** intersection method, inaccessible point, geocentric coordinates, Earth Gravity Model, deflection of the vertical, Gauss–Helmert method

## Abstract

Surveyors are occasionally tasked to with determining the coordinates of inaccessible locations or points in civil engineering applications, ground control points for photogrammetry or LiDAR data acquisition, among others. The present work outlines and investigates a novel method for estimating the GNSS coordinates of an inaccessible location where a surveying instrument cannot be set up. The procedure is based on the well-known surveying intersection method and data extracted from an Earth Gravity Model (e.g., EGM 2008). The location’s coordinates are obtained from the least-squares adjustment of the angles and distances measured from at least two sites to the unknown point using a total station, within the framework of the Gauss–Helmert method. Field tests confirmed that the accuracy of the determined coordinates of the inaccessible point is at the level of 1 cm. The proposed method bypasses standard coordinate transformation steps performed with the traditional approach, directly producing geocentric coordinates of the unknown points.

## 1. Introduction

Determination of the three-dimensional coordinates of locations or unknown points is a fundamental task in professional surveying. Various measurement techniques can be used for this purpose, among which the GNSS (Global Navigation Satellite System) method is probably the most obvious. However, the GNSS method cannot be used if the GNSS receiver is obstructed (e.g., by tall buildings in the city canyons) or if the location is inaccessible. Nevertheless, the goal is still achievable if the inaccessible point is visible from at least two points using a GNSS station, and a total station is on hand to measure angles and distances to the unknown point from the GNSS accessible points.

The problem of integrating various geodetic data types has been an active research topic for a long time. Some relevant publications on this topic include, e.g., [1,2,3,4,5]. These publications demonstrate different approaches to combining various spatial data types with joint adjustment and visualisation. The authors analysed different GNSS networks, which were a base for other measurements (for example, total station) and for capturing inaccessible points. The results are also used in engineering tasks (like deformation monitoring) or for providing control surveys (e.g., rock block displacements) in the mountains. Data integration and fusion procedures were presented regarding integrating available geoid models. Then, the objects were spatially analysed and adjusted in a coherent, unique reference system, giving more satisfactory results than by processing the data separately. Implementing combined surveying methods, especially total station and GNSS methods requires continuous scientific research and validation under field conditions. Moreover, similar approaches can successfully be utilised in the mining industry [6]. Examples of compelling applications and reports on developing spatial data integration methodologies are published in [7]—both cases concern deformation surveys and precise levelling based on the joint adjustment of multi-source data. In the case of open pit mines, one can observe a significant influence of atmospheric refraction and, at long distances, the Earth’s curvature (plumb-line deviations). Similar problems are also visible when performing levelling across wide rivers. In this case, geodetic works must establish special control networks on two opposite riverbanks and utilise unique instrumental designs (e.g., integrated prisms aided by LED lighting). Today, such measurements can be performed with millimetre-level accuracies.

However, such cases require a skilful fusion of the mentioned aspects, supported by advanced computational methods. In addition to GNSS and total station measurements, other technologies like laser scanning or photogrammetry are also worth noting. Some examples of utilising the above surveys are presented in [8,9]. The first paper focuses on the issue of inventory surveying of façade scaffolding. The authors considered various land surveying methods such as photogrammetry, laser scanning or total station. The developed method for integrating the abovementioned data allows for the effective monitoring of scaffolding on construction sites. In the second article, the authors demonstrate their newly developed calculation and data adjustment method for precise object dimensioning by employing close-range photogrammetry. This solution utilises a modified Gauss–Markov approach to adjust the spatial control network deployed directly on the surveyed object. Both demonstrated cases offer singular millimetre accuracies. In modern engineering geodesy, the development of computational methods—especially planimetric and numerical—has become particularly important.

The spatial intersection method, well known to surveyors, allows for the determination of three-dimensional point coordinates based on angle and distance measurements from at least two points where total station/GNSS receivers can be placed (e.g., [10,11,12,13]). In addition, in [14], the authors examined the accuracy of coordinates of inaccessible points determined using the intersection method. Such an approach can be successfully employed where measuring direct distances is impossible. Hence, it finds numerous applications in civil engineering tasks, including the setting out of control networks, control surveys or similar jobs [15]. In [16], however, a novel concept in modern geodesy, the Gauss–Helmert model (GHM), was used. The GHM [17] is a variation of the least-squares method designed to minimise errors in all variables, including parameters and observations. It is formulated as linear conditional equations with unknown parameters. The proposed intersection method for the coordinate determination of an inaccessible location uses linear conditional equations. Hence, the GHM is a relevant approach to solving the system of observation equations.

Textbooks provide in-depth background information on this approach, e.g., [18]. In addition, the properties of this method have also been analysed in the literature, e.g., [19,20,21,22]. Various authors have also studied related issues, including similarity transformation problems under the GHM, proposed alternative solutions [23,24], and compared with other similar approaches, e.g., Gauss–Newton [25]. Also, some more advanced topics, including the GHM with a singular dispersion matrix [26], parameter estimation within the GHM using the Pareto optimality [27] or robust parameter estimation of the non-linear GHM [28], have been investigated.

The present paper introduces the GHM for the intersection method. The proposed approach integrates GNSS observations and the total station’s deflections of the vertical axis, determined based on a global Earth Gravity Model (EGM) [29]. The authors’ primary motivation is to derive the most general GHM of observational equations binding various possible measurements in the proposed spatial intersection. Moreover, one of the leading practical aspects was the ability to offer an efficient method for surveying inaccessible points in measuring services employing real-time GNSS receivers. A similar approach was already tested and published in [30]; however, the demonstrated technique has been completed using the GHM. Above all, it should be noted that the classic surveys, together with their modern forms using, e.g., aerial or terrestrial laser scanning, can be reliably integrated and combined with GNSS by expressing the results in a common geocentric datum. Thus, we introduced the geocentric intersection method, which includes rigorous accuracy assessment.

The specific processing steps that are performed in the present paper can be synthesized as follows:The problem: to obtain precise geocentric coordinates of inaccessible locations for photogrammetry or LiDAR survey.A method is proposed based on the directional intersection of the inaccessible locations using a total station from two stations with measured GNSS coordinates.Solution: the non-linear directional intersection observation equations are formed.The equally weighted least-squares solution of the non-linear observational equations (LMA) is calculated.Linearisation of observation equations and formulation of the Gauss–Helmert model (GHM) is carried out.A solution of the GHM (the weighted least-squares method) is obtained.Field experiments are carried out. The results of the LMA and GHM are compared.

## 2. Materials and Methods

### 2.1. Method

#### 2.1.1. Basic Relationships

Using a total station, one can measure the horizontal *α* and vertical *β* angles and distance *s* from point **P** to target **Q** (Figure 1). Assuming that the vertical angle *β* is corrected due to the atmospheric refraction and the instrument/mirror height (*i*)/(*j*) is considered, the coordinates (x, y, z) of target **Q** are computed from (1):(1)$$x=\left(\begin{array}{l} x\\ y\\ z \end{array}\right)=\left(\begin{array}{l} s\cos\alpha \sin\beta \\ s\sin\alpha \sin\beta \\ s\cos\beta +i-j \end{array}\right)$$

The coordinates **x** of target **Q** are referenced to a local coordinate system (x, y, z). They must be transformed into the destination reference system. As a destination reference system, we use the geocentric World Geodetic System 1984 (WGS84), which the GNSS uses.

Referring to Figure 1, the origin of the local reference system is located at **P**. Point **P** with geocentric coordinates (X, Y, Z) in the GNSS reference system and the geodetic latitude *φ* and longitude *λ*. The orientation of the local system (x, y, z) in the geocentric system (X, Y, Z) is defined by the three orientation angles: *Σ*, *ξ*, *η*. The so-called total station horizontal orientation constant is represented by the angle *Σ*. The *ξ* and *η* are vertical deflection angle components (north/east) from the normal to the ellipsoid at **P**.

The coordinates of the point Q in the local system (x, y, z) can be transformed into geocentric coordinates (X′, Y′, Z′) using (2) (see, e.g., [3,31]):(2)$$\left(\begin{array}{l} X^{\prime }\\ Y^{\prime }\\ Z^{\prime } \end{array}\right)=\left(\begin{array}{l} X\\ Y\\ Z \end{array}\right)+{\left[R\left(\Sigma \right)\cdot Q\left(\xi ,\eta ,\varphi \right)\cdot P\left(\varphi ,\lambda \right)\right]}^{T}\left(\begin{array}{l} x\\ y\\ z \end{array}\right)$$
or in a shorter form using (3)
(3)$$X^{\prime }=X+{\left(RQP\right)}^{T}x$$
where the matrices **P**, **Q**, and **R** are given by (4)–(6):(4)$$P\left(\varphi ,\lambda \right)=\left(\begin{array}{ccc} -\sin\varphi \cos\lambda & -\sin\varphi \sin\lambda & \cos\varphi \\ -\sin\lambda & \cos\lambda & 0\\ \cos\varphi \cos\lambda & \cos\varphi \sin\lambda & \sin\varphi \end{array}\right)$$
(5)$$Q\left(\xi ,\eta ,\varphi \right)=\left(\begin{array}{ccc} 1 & -\eta \tan\varphi & -\xi \\ \eta \tan\varphi & 1 & -\eta \\ \xi & \eta & 1 \end{array}\right)$$
(6)$$R\left(\Sigma \right)=\left(\begin{array}{ccc} \cos\Sigma & \sin\Sigma & 0\\ -\sin\Sigma & \cos\Sigma & 0\\ 0 & 0 & 1 \end{array}\right)$$

The geocentric coordinates of P(X, Y, Z), together with an Earth Gravity Model, e.g., EGM 2008, allow for the computation of the components (ξ, η) of the deflection of the vertical (see Section 2.3 point 3 for more details). The orientation parameter Σ can be determined by solving Equation (2) for a given set of observations (*s*, *α*, *β*, *i*, *j*), GNSS coordinates of the station P(X, Y, Z) and the target Q(X′, Y′, Z′). For instance, the Levenberg–Marquardt [32] method can be used.

**Figure 1 sensors-23-08199-f001:**
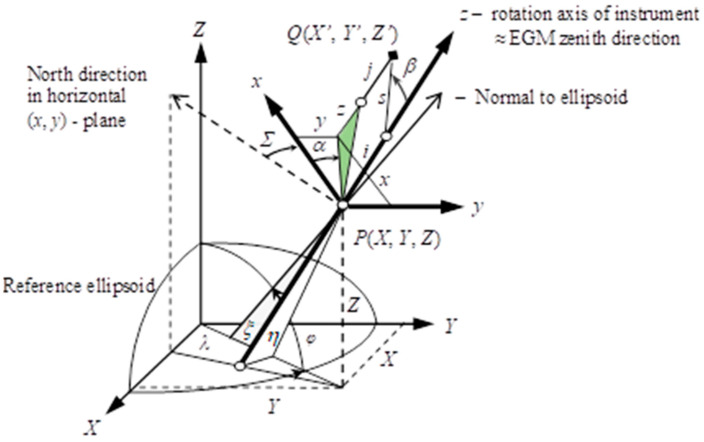
Relationships between the instrument and the geocentric reference system. Source: [33].

#### 2.1.2. Least-Squares Solution of the Intersection Equations

Referring to Figure 2a and Equation (3), a system of non-linear equations binding the observations (s_12_, α_12_, β_12_, s_21_, α_21_, β_21_, α_1A_, β_1A_, α_2A_, β_2A_, ξ_1_, η_1_, ξ_2_, η_2_, X_1_, X_2_) and the parameters (s_1A_, Σ_2A_, Σ_1_, Σ_2_,) can be formulated. For the four measured angles and distances between **1** and **2**, we have (7) and (8) (see, e.g., [34,35]):
(7)$$X_{2}=X_{1}+{\left[R\left(\Sigma_{1}\right)\cdot Q\left(\xi_{1},\eta_{1},\varphi_{1}\right)\cdot P\left(\varphi_{1},\lambda_{1}\right)\right]}^{T}x(s_{12},\alpha_{12},\beta_{12},i_{1},j_{2})$$
(8)$$X_{1}=X_{2}+{\left[R\left(\Sigma_{2}\right)\cdot Q\left(\xi_{2},\eta_{2},\varphi_{2}\right)\cdot P\left(\varphi_{2},\lambda_{2}\right)\right]}^{T}x(s_{21},\alpha_{21},\beta_{21},i_{2},j_{1})$$

The coordinates of the target **A** can be calculated using the set of observations obtained at **1**, independently of observations obtained at **2**:(9)$$X_{A}=X_{1}+{\left[R\left(\Sigma_{1}\right)\cdot Q\left(\xi_{1},\eta_{1},\varphi_{1}\right)\cdot P\left(\varphi_{1},\lambda_{1}\right)\right]}^{T}x(s_{1A},\alpha_{1A},\beta_{1A},i_{1A},j_{1A})$$
(10)$$X_{A}=X_{2}+{\left[R\left(\Sigma_{2}\right)\cdot Q\left(\xi_{2},\eta_{2},\varphi_{2}\right)\cdot P\left(\varphi_{2},\lambda_{2}\right)\right]}^{T}x(s_{2A},\alpha_{2A},\beta_{2A},i_{2A},j_{2A})$$

Since (9) and (10) should be equal, a conditional equation is expressed by (11):(11)$$\begin{array}{l} X_{2}-X_{1}={\left[R\left(\Sigma_{1}\right)\cdot Q\left(\xi_{1},\eta_{1},\varphi_{1}\right)\cdot P\left(\varphi_{1},\lambda_{1}\right)\right]}^{T}x(s_{1A},\alpha_{1A},\beta_{1A},i_{1},j_{A})\\ \;\;\;-{\left[R\left(\Sigma_{2}\right)\cdot Q\left(\xi_{2},\eta_{2},\varphi_{2}\right)\cdot P\left(\varphi_{2},\lambda_{2}\right)\right]}^{T}x(s_{2A},\alpha_{2A},\beta_{2A},i_{2},j_{A}) \end{array}$$

Equations (7), (8) and (11) constitute a system of nine independent, non-linear equations with four unknowns (s_1A_, s_2A_, Σ_1_, Σ_2_). The Levenberg–Marquardt algorithm (LMA) can be utilised to estimate unknowns. Finally, the coordinates of target A are computed from Equations (9) and (10).

#### 2.1.3. Differential Relationships

A linear set of equations is required to perform the LSQ adjustment. The linear version of the equations is achieved by expanding (2) into the Taylor series. It can be achieved using the differential correction d**X′** to the position **X′** of the point **Q** (Figure 1) given by:(12)$$\begin{array}{l} dX^{\prime }=dX+{\left(\frac{dR}{d\Sigma }\cdot Q\cdot P\right)}^{T}xd\Sigma +{\left(R\cdot \frac{dQ}{d\xi }\cdot P\right)}^{T}xd\xi +{\left(R\cdot \frac{dQ}{d\eta }\cdot P\right)}^{T}xd\eta \;\\ \;\;+{\left(R\cdot Q\cdot P\right)}^{T}(\frac{dx}{ds}ds+\frac{dx}{d\alpha }d\alpha +\frac{dx}{d\beta }d\beta ) \end{array}$$
where the partial derivatives of Equations (1), (5) and (6) are given by (13)–(15):(13)$$\frac{dR}{d\Sigma }=\left[\begin{array}{ccc} -\sin\Sigma & \cos\Sigma & 0\\ -\cos\Sigma & -\sin\Sigma & 0\\ 0 & 0 & 0 \end{array}\right]$$
(14)$$\left[\begin{array}{cc} \frac{dQ}{d\xi } & \frac{dQ}{d\eta } \end{array}\right]=\left[\begin{array}{cc} \begin{array}{ccc} 0 & 0 & -1\\ 0 & 0 & 0\\ 1 & 0 & 0 \end{array} & \begin{array}{ccc} 0 & -\tan\varphi & 0\\ \tan\varphi & 0 & -1\\ 0 & 1 & 0 \end{array} \end{array}\right]$$
(15)$$\left[\begin{array}{ccc} \frac{dx}{ds} & \frac{dx}{d\alpha } & \frac{dx}{d\beta } \end{array}\right]=\left[\begin{array}{lll} \cos\alpha \sin\beta & -s\sin\alpha \sin\beta & s\cos\alpha \cos\beta \\ \sin\alpha \sin\beta & s\cos\alpha \sin\beta & s\sin\alpha \cos\beta \\ \cos\beta & 0 & -s\sin\beta \end{array}\right]$$
where
(16)$$A_{\Sigma }={\left(\frac{dR}{d\Sigma }\cdot Q\cdot P\right)}^{T}x,A_{\xi }={\left(R\cdot \frac{dQ}{d\xi }\cdot P\right)}^{T}x,A_{\eta }={\left(R\cdot \frac{dQ}{d\eta }\cdot P\right)}^{T}x,$$
(17)$$A_{s}={\left(R\cdot Q\cdot P\right)}^{T}\frac{dx}{ds},A_{\alpha }={\left(R\cdot Q\cdot P\right)}^{T}\frac{dx}{d\alpha },A_{\beta }={\left(R\cdot Q\cdot P\right)}^{T}\frac{dx}{d\beta }$$
(18)$$\Delta X_{P,Q}={\left(R_{P}Q_{P}P_{P}\right)}^{T}x_{P,Q}$$

Finally, Equation (12) can be written as (19):(19)$$dX^{\prime }=dX+A_{\Sigma }d\Sigma +A_{\xi }d\xi +A_{\eta }d\eta +A_{s}ds+A_{\alpha }d\alpha +A_{\beta }d\beta$$

The differential Equation (19) is used for setting up the linear mixed-observation Gauss–Helmert model of the spatial intersection.

#### 2.1.4. LSQ Adjustment of the Intersection Based on the Gauss–Helmert Model

The non-linear LSQ solution of the intersection equations presented in Section 3.1 can be improved by considering the standard deviations of the measurements. By adding small corrections to the measured quantities as well as to the approximate values of the unknown parameters, the fundamental non-linear Equation (2) or (3) can be written in the form:(20)$$\begin{array}{l} X^{\prime }+dX^{\prime }=X+dX+\\ +{\left(R\left(\Sigma +d\Sigma \right)\cdot Q\left(\xi +d\xi ,\eta +d\eta ,\varphi \right)\cdot P\left(\varphi ,\lambda \right)\right)}^{T}x(s+ds,\alpha +d\alpha ,\beta +d\beta ,i,j) \end{array}$$

Then, using Equation (19), we obtain the linear form of Equation (20):(21)$$\begin{array}{l} X^{\prime }+dX^{\prime }=X+dX+\Delta X_{PQ}+\\ +A_{\Sigma }d\Sigma +A_{\xi }d\xi +A_{\eta }d\eta +A_{s}ds+A_{\alpha }d\alpha +A_{\beta }d\beta \end{array}$$
with
(22)$$\Delta X_{PQ}={\left(R_{P}Q_{P}P_{P}\right)}^{T}x_{PQ}$$

Therefore, the non-linear Equations (7), (8) and (11) can be written in linear form as (23)–(25):(23)$$\begin{array}{l} X_{2}+v_{X_{2}}=X_{1}+v_{X_{1}}+\Delta X_{12}+\\ +A_{\Sigma_{12}}d\Sigma_{1}+A_{\xi_{12}}v_{\xi_{1}}+A_{\eta_{12}}v_{\eta_{1}}+A_{s_{12}}v_{s_{12}}+A_{\alpha_{12}}v_{\alpha_{12}}+A_{\beta_{12}}v_{\beta_{12}} \end{array}$$
(24)$$\begin{array}{l} X_{1}+v_{X_{1}}=X_{2}+v_{X_{2}}+\Delta X_{21}+\\ +A_{\Sigma_{21}}d\Sigma_{2}+A_{\xi_{21}}v_{\xi_{2}}+A_{\eta_{21}}v_{\eta_{2}}+A_{s_{21}}v_{s_{21}}+A_{\alpha_{21}}v_{\alpha_{21}}+A_{\beta_{21}}v_{\beta_{21}} \end{array}$$
(25)$$\begin{array}{l} X_{2}+v_{X_{2}}+\Delta X_{2A}+A_{\Sigma_{2A}}d\Sigma_{2}+A_{\xi_{2A}}v_{\xi_{2}}+A_{\eta_{2A}}v_{\eta_{2}}+A_{s_{2A}}ds_{2A}+A_{\alpha_{2A}}v_{\alpha_{2A}}+A_{\beta_{2A}}v_{\beta_{2A}}\\ =X_{1}+v_{X_{1}}+\Delta X_{1A}+A_{\Sigma_{1A}}d\Sigma_{1}+A_{\xi_{1A}}v_{\xi_{1}}+A_{\eta_{1A}}v_{\eta_{1}}+A_{s_{1A}}ds_{1A}+A_{\alpha_{1A}}v_{\alpha_{1A}}+A_{\beta_{1A}}v_{\beta_{1A}} \end{array}$$
where v_s_, v_α_ and v_β_ are random errors of the distance and angles *s*, *α*, *β*, respectively,

v_ξ_, v_η_ are random errors of the deflection of the vertical components, *ξ*, *η*,

*dΣ* and *ds* are corrections to the approximate values of the azimuth *Σ* and distance *s*, and

**v_x_** = (v_x_, v_Y_, v_Z_)^T^ is the vector of random errors of the GNSS coordinates.

The system of three matrix–vector observation equations (Equations (23)–(25)) can be written as the Gauss–Helmert model in the matrix form (26):(26)Bv+Ap+w=0
where
(27)$$B=\left[\begin{array}{cccccccccccccccc} A_{s_{12}} & A_{\alpha_{12}} & A_{\beta_{12}} & 0 & 0 & 0 & 0 & 0 & 0 & 0 & A_{\xi_{12}} & A_{\eta_{12}} & 0 & 0 & I & -I\\ 0 & 0 & 0 & A_{s_{21}} & A_{\alpha_{21}} & A_{\beta_{21}} & 0 & 0 & 0 & 0 & 0 & 0 & A_{\xi_{21}} & A_{\eta_{21}} & -I & I\\ 0 & 0 & 0 & 0 & 0 & 0 & A_{\alpha_{1A}} & A_{\beta_{1A}} & -A_{\alpha_{2A}} & -A_{\beta_{2A}} & A_{\xi_{1A}} & A_{\eta_{1A}} & -A_{\xi_{2A}} & -A_{\eta_{2A}} & I & -I \end{array}\right]$$
(28)$$A=\left[\begin{array}{cccc} A_{\Sigma_{12}} & 0 & 0 & 0\\ 0 & A_{\Sigma_{21}} & 0 & 0\\ A_{\Sigma_{1A}} & -A_{\Sigma_{2A}} & A_{s_{1A}} & -A_{s_{2A}} \end{array}\right]$$
(29)$$w=\left[\begin{array}{c} X_{1}+\Delta X_{1,2}-X_{2}\\ X_{2}+\Delta X_{2,1}-X_{1}\\ X_{1}+\Delta X_{1,A}-X_{2}-\Delta X_{2,A} \end{array}\right]$$
and

$v={\left[v_{s_{12}},v_{\alpha_{12}},v_{\beta_{12}},v_{s_{21}},v_{\alpha_{21}},v_{\beta_{21}},v_{\alpha_{1A}},v_{\beta_{1A}},v_{\alpha_{2A}},\;v_{\beta_{2A}},v_{\xi_{1}},v_{\eta_{1}},\;v_{\xi_{2}},v_{\eta_{2}},v_{X_{1}}^{T},v_{X_{2}}^{T}\right]}^{T}$ is the vector of random errors of the observations,

**I** is the 3 × 3 identity matrix, and

$p={\left[d\Sigma_{1},d\Sigma_{2},ds_{1A},ds_{2A}\right]}^{T}$ represents the corrections to approximate values of the

unknowns $p_{0}={\left[\Sigma_{1},\Sigma_{2},s_{1A},s_{2A}\right]}^{T}$.

The Gauss–Helmert model assumes condition (30):(30)$$v^{T}Pv=\min$$
where

$P=\Sigma_{l_{b}}^{-1}$ is the weight matrix of the observations,

$l_{b}={\left[s_{12},\alpha_{12},\beta_{12},s_{12},\alpha_{12},\beta_{12},\alpha_{1A},\beta_{1A},\alpha_{2A},\beta_{2A},\xi_{1},\eta_{1},\xi_{2},\eta_{2},X_{1}^{T},X_{2}^{T}\right]}^{T}$, and

$\Sigma_{l_{b}}^{}$ is the covariance matrix.

According to the traditional Lagrange approach, the target function consisting of (30) and the linear combination (Lagrange multipliers) of condition Equation (26) is formed. The solution using Euler–Lagrange conditions (partial derivatives equal to zero) yields the following (see, e.g., [36]):(31)$$p=-{\left(A^{T}M^{-1}A\right)}^{-1}A^{T}M^{-1}w$$
(32)$$v=-P^{-1}B^{T}M^{-1}(Ap+w)$$
where $M=BP^{-1}B^{T}$

Hence, the parameters’ adjusted observations and their covariance matrices are:(33)$$p_{a}=p_{0}+p$$
(34)$$l_{a}=l_{b}+v$$
(35)$$\Sigma_{p_{a}}={\left(A^{T}M^{-1}A\right)}^{-1}$$
(36)$$\Sigma_{l_{a}}=\Sigma_{l_{b}}-\Sigma_{v}$$

The covariance matrix of residual vector **v** is given by (37):(37)$$\Sigma_{v}=-P^{-1}B^{T}M^{-1}[M-A{\left(A^{T}M^{-1}A\right)}^{-1}A^{T}]M^{-1}BP^{-1}$$

The corrections to the observations **v** are acceptable if they fulfil the local test as follows (38):(38)$$max\left\{\frac{\left|v_{i}\right|}{\sigma_{v_{i}}}\right\}\leq 3$$
where *i* = *s*, *α*, *β*, X, Y, Z, *ξ*, *η* and σ_v_ is the standard deviation of correction **v**.

The GHM-based observation equation model is developed here to adjust the observations and parameters found in the vector **p** according to the LSQ method.

### 2.2. Experimental Design

Figure 2 shows the survey design of the experiments. The total station and the GNSS receiver were positioned at points **1** and **2** in the first experiment and at points **3** and **4** during the second experiment. Target **A** or targets **B**, **C** and **D** were located on the roof of the building some 18 m above the level of stations **1**–**4**. The approximate distance between the instrument positions and the targets was determined using existing large-scale maps. At each station, the horizontal and vertical angles and distances were measured using a total station. The instrument height (*i*) and the mirror at the target were also measured. In the next step, RTK GNSS observations were taken at all points, including the target **A**–**D** points. The GNSS coordinates of the points (**A**–**D**) and the distances to them were considered as the controls of the experiments, and as such, they were not included in the calculations of the GNSS coordinates of the target points.

### 2.3. Data

Both experiments used a Leica Nova MS50 total station to measure the angles and distances (*s*, *α*, *β*). The heights of the instrument (*i*) and reflector (*j*) were also measured. The measurements obtained during the first experiment are listed in Table 1. The symbol ”**A**” denotes an inaccessible point, while **1** and **2** denote sites of the total stations from which measurements were carried out.The geocentric coordinates (*X*, *Y*, *Z*) of points **1** and **2** were measured using the Leica Viva GS08plus SmartAntenna GNSS receiver. The ASG-EUPOS service (https://www.asgeupos.pl/index.php, accessed on 25 September 2023) provided the RTK correction. The geocentric GNSS coordinates are listed in Table 2.The north-south *ξ* and east-west *η* components of the deflection of the vertical axis of the instrument, relative to the normal to ellipsoid, can be computed from an Earth Gravity Model, for example, EGM 2008 (https://www.usna.edu/Users/oceano/pguth/md_help/html/egm96.htm, accessed on 25 September 2023). The computations are based on the fundamental relations of the physical geodesy:
*ξ* = *Φ* − *φ*,(39)
*η* = (*Λ* − *λ*) *cos φ*(40)where*φ* and *λ* are the latitude and longitude obtained from GNSS, and*Φ* and *Λ* are the astronomical latitude and longitude, respectively.

The astronomical latitude *Φ* and longitude *Λ* are computed from the Earth Gravity Model (EGM 2008), following equations outlined in [37,38,39], i.e., $\Phi =\arctan((-\partial W/\partial Z)/\sqrt{{\left(\partial W/\partial X\right)}^{2}+{\left(\partial W/\partial Y\right)}^{2}}$ and Λ=arctan((∂W/∂Y)/(∂W/∂X), where *W* is the potential of the gravity field in the EGM 2008 geoid. Here, this method was used to calculate the deflection of vertical components.

An alternative way to obtain the vertical deflection components is the global vertical deflection (GVD) model available from the US National Geospatial Intelligence Agency (NGA) [40]. The calculated deflection of vertical components is listed in Table 3.

Table 4 and Table 5 show the observations collected while executing Experiment 2.

Both experiments were controlled by GNSS measurements carried out at locations assumed to be inaccessible, i.e., **A**–**D**. These measurements were not a part of adjusting the observations.

## 3. Results

### 3.1. Experiment 1

#### 3.1.1. The Levenberg–Marquardt Algorithm (LMA) Adjustment

We use the LMA to solve non-linear Equations (7), (8) and (11). The initial values of unknowns, i.e., distances *s*(**1**–**A**) = 43.1 m, s(**2**–**A**) = 40.3 m and horizontal orientation angles *Σ*_1_ = 70 gon, *Σ*_2_ = 200 gon were obtained from a large-scale map. The adjusted unknowns were subsequently used in Equations (9) and (10) to calculate the coordinates of target **A**. Table 6 shows the results.

The differences between the adjusted unknowns *s*(**1**–**A**) and *s*(**2**–**A**) and the control measurements (Table 1) equal 0.001 m and 0.021 m, respectively, while the differences between the calculated versus measured GNSS coordinates of the target A (Table 2) are −0.004 m, 0.007 m and −0.002 m. These differences are significantly smaller than three times their standard deviations.

#### 3.1.2. The Gauss–Helmert Model (GHM)

The Gauss–Helmert model is used for the rigorous LSQ observations’ adjustment. Unlike the LMA, it considers a priori input data accuracy, which is essential if observations are of varying precision.

We adjusted the observations using both methods to compare the LMA and GHM solutions. For the approximate values of unknowns, the LMA solution was used (Table 6). These values were also used to compute matrices **A** and **B** and the vector **w** according to Equations (27)–(29). Subsequently, we received the following solution for the unknown, as shown in Table 7.

The GHM produced more accurate results than the LMA because the differences between the unknowns *s*(**1**–**A**) and *s*(**2**–**A**) and the controls are now 0.005 m and 0.013 m, respectively, vs. 0.001 m and 0.021 m as per the LMA solution.

This method produces more consistent results because the GHM involves all observations, coordinates of stations **1** and **2** and parameters in the adjustment.

Table 7 shows point **A**’s geocentric coordinates computed from Equations (9) and (10). The coordinates vary by 0.002 m, −0.002 m and 0.001 m from the GNSS control for X, Y and Z, respectively.

The coordinate differences of target **A** derived using the LMA and GHM are shown in Table 8. Evidently, the GHM method outperforms the LMA algorithm.

The distance and coordinate differences are less than three times their standard deviations. The standard deviations of the computed coordinates of target **A** are 0.007 m, 0.013 m and 0.008 m, respectively.

The ratio of corrections of adjusted observation **v** and their standard deviations is 2.8, indicating that the local test—Equation (38)—has been fulfilled.

### 3.2. Experiment 2

Experiment 2 was conducted to further verify the GHM adjustment’s usefulness in the proposed computational method for determining the precise coordinates of inaccessible locations under different GNSS satellite deployments and likely different ionosphere and troposphere conditions. The GNSS coordinates of three points, **B**, **C** and **D** (Figure 2b), were determined in the test.

The results show that the computed coordinates are consistent with GNSS control coordinates. The coordinate differences are below 1 cm (except for one) and are within one standard deviation of the coordinates. Also, the root-mean-square error (RMSE) calculated for the coordinate differences is below 1 cm (Table 9).

## 4. Discussion

The proposed method for determining the coordinates of inaccessible locations uses two orientation parameters (*ξ*, *η*), calculated for the stations’ points from EGM 2008. Hence, the parameters link the total station measurements and the gravity field. Due to this feature, the approach allows for precise, multi-technique direct positioning in the global reference frame. The availability of the global gravity model EGM 2008 allows for straightforward calculations of (*ξ*, *η*), following Equations (39) and (40) [41]. However, to test the impact on the adjusted coordinates of the vertical deflection, we solved the case when the vertical deflection equals *ξ* = 0, *η* = 0. In this case, the rotation axis of the instrument is perpendicular to the ellipsoid. Hence, the rotation matrix **Q** = **I**, δ**Q**/δξ = **0**, δ**Q**/δη = **0**, **A**_ξ_ = **0**, **A**_η_ = **0**, meaning that the adjustment problem is simplified. We obtained target **A**’s estimated coordinates as shown in Table 10.

There is a little difference (0.002 m) in the coordinates between the simplified case (*ξ* = 0, *η* = 0) and the actual situation when, typically, *ξ* ≠ 0, *η* ≠ 0. The simplified solution can introduce systematic errors in the coordinates, particularly in mountainous areas or when aiming at the target requires high vertical angles. Hence, the simplified procedure is not recommended in the discussed cases.

The coordinate differences between the non-linear LMA and the GHM solutions are minor in our tests. However, the results may significantly vary in another measurement scenario (different geometry of the intersection, more/less accurate instrument) because the LMA provides a non-weighted solution compared with the GHM-weighted solution. Also, the GHM approach provides a rigorous accuracy assessment of the solution.

## 5. Conclusions

This paper introduced a method for positioning inaccessible points using the novel DIM method directly in the geocentric reference frame. This solution is based on the LSQ adjustment of measurements and parameter estimation, which includes conditional equations with unknowns within the Gauss–Helmert model. The method integrates total station observations, GNSS positioning, and vertical-axis deflection components.

The proposed method is a new contribution to precise multi-technique surveying [42]. The field tests showed that the proposed method allows for the determination of the coordinates of inaccessible locations in the geocentric reference frame at the accuracy of approximately 1 cm in terms of the standard deviation of point positioning. Many industries using these results can benefit from such accuracies. In particular, a high level of accuracy is required when monitoring the displacement and deformation of engineering structures. It can be used for engineering surveying jobs and for establishing ground control points for LiDAR and aerial photograph acquisition.

Observing the current development trends in surveying, characterised by the rapid development of modern instruments, the demand for efficient real-time or near-real-time data fusion methods is growing. These methods are vital for laser scanning and photogrammetry for object dimensioning or structural monitoring.

The proposed method is innovative, and thanks to it, one can significantly increase the accuracy and effectiveness of the job execution, mainly when a job must be performed in real or near-real time. We note such cases concerning the still-growing scope of deformation monitoring offered to construction contractors. In addition, as quantity surveying practice indicates, conducting measurement work in usually challenging construction conditions often makes it impossible to observe some control points. Such situations demand the surveyor’s flexibility, and spontaneous decisions may often change the geometry of the previously planned control network in order to adapt to sudden field situations. Our proposed approach significantly reduces such risks, offering a flexible method for capturing points efficiently while keeping the highest accuracy. Future planned works include the continued development of computational methods for precise point positioning. Notably, this is important in the deformation and displacement monitoring of hazard sites near intensive industrial exploitation areas.

## Figures and Tables

**Figure 2 sensors-23-08199-f002:**
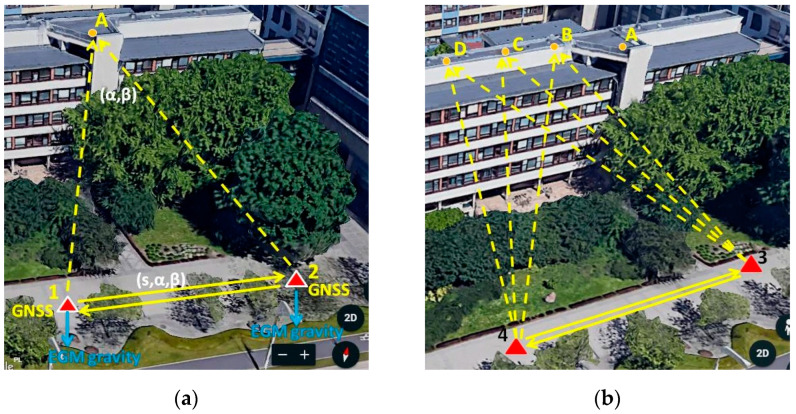
Survey design of (**a**) Experiment 1 and (**b**) Experiment 2. The GNSS receiver occupied all points **1**–**4** and **A**–**D**. The total station was positioned at stations 1–4. The instrument (*i*) and mirror (*j*) heights were also measured and included in the calculations.

**Table 1 sensors-23-08199-t001:** The data for Experiment 1. **1** and **2** denote instrument sites, and **A** is the location to be determined. Refer to Figure 1 and Figure 2a for more details.

Observation	*s* (m)	*α* (gon)	*β* (gon)	*i* (m)	*j* (m)
1 to 2	37.121	0.0489	100.1286	1.611	1.500
1 to A	43.571	339.2618	65.1532	1.611	2.150
2 to 1	37.124	71.5203	100.2894	1.635	1.500
2 to A	40.953	141.2695	62.7610	1.635	2.150
Std Dev.	0.006	0.001	0.001	0.001	0.001

**Table 2 sensors-23-08199-t002:** The geocentric GNSS coordinates for Experiment 1 (Figure 1 and Figure 2a). Note that the coordinates of point **A** were measured as the control for the experiment.

Coordinate	1	2	A	Std Dev.
X (m)	3,835,779.346	3,835,758.231	3,835,763.321	0.008
Y (m)	1,177,321.994	1,177,351.033	1,177,324.809	0.008
Z (m)	4,941,536.189	4,941,545.624	4,941,576.310	0.008

**Table 3 sensors-23-08199-t003:** Calculated deflection of vertical components.

Deflection of Vertical	1	2	Std Dev.
*ξ* (″)	5.9926	5.9852	1
*η* (″)	6.2033	6.1967	1

**Table 4 sensors-23-08199-t004:** The total station measurement data for the testing points B, C and D (Figure 2b).

Observation	*s* (m)	*α* (gon)	*β* (gon)	*i* (m)	*j* (m)
3 to 4	33.007	31.8735	100.4359	1.685	1.500
3 to B	40.753	107.2129	63.2411	1.685	1.900
3 to C	43.474	94.5445	65.8200	1.685	1.900
3 to D	47.413	85.4091	68.9707	1.685	1.900
4 to 3	33.009	225.3364	100.2523	1.657	1.500
4 to B	43.612	161.3899	65.8203	1.657	1.900
4 to C	40.228	149.3560	62.6006	1.657	1.900
4 to D	38.690	135.5829	60.9652	1.657	1.900
Std Dev.	0.006	0.001	0.001	0.001	0.001

**Table 5 sensors-23-08199-t005:** The geocentric coordinates for Experiment 2 points **B**, **C** and **D** (Figure 2b). Note that the GNSS coordinates of points **B**, **C** and **D** were measured as controls only.

Point	3	4	B	C	D	Std Dev.
**X** (m)	3,835,762.327	3,835,780.698	3,835,764.596	3,835,769.196	3,835,773.170	0.008
**Y** (m)	1,177,338.201	1,177,311.976	1,177,313.716	1,177,307.830	1,177,302.003	0.008
**Z** (m)	4,941,545.590	4,941,537.594	4,941,577.938	4,941,575.760	4,941,574.056	0.008

**Table 6 sensors-23-08199-t006:** Adjusted unknowns (distances and horizontal angles) and coordinates of target **A**.

Unknown	*s*(1–A)	*s*(2–A)	*Σ* _1_	*Σ* _2_
	43.572 m	40.974 m	73.4638 gon	201.9942 gon
**Coordinates** of **A**				
**X** (m)	3,835,763.325			
**Y** (m)	1,177,324.803			
**Z** (m)	4,941,576.312			

**Table 7 sensors-23-08199-t007:** The adjusted unknowns and coordinates of the inaccessible target **A** using the GHM model.

Unknown	*s*(1–A)	*s*(2–A)	*Σ* _1_	*Σ* _2_
	43.576 m	40.966 m	73.4693 gon	201.9980 gon
**Coordinates** of **A**				
**X** (m)	3,835,763.322			
**Y** (m)	1,177,324.807			
**Z** (m)	4,941,576.311			

**Table 8 sensors-23-08199-t008:** Coordinate differences of the inaccessible target **A** calculated using the LMA and GHM models (LMA/GHM result *minus* GNSS-control).

Model	d_X_ (m)	d_y_ (m)	d_z_ (m)
LMA	−0.004	0.007	−0.002
GHM	0.002	−0.002	0.001

**Table 9 sensors-23-08199-t009:** Coordinate differences (computed *minus* GNSS control) for targets **B**, **C** and **D**.

Target	*δX* (m)	*δY* (m)	*δZ* (m)	$\sqrt{{\delta X}^{2}+{\delta Y}^{2}+{\delta Z}^{2}}$ (m)
**B**	−0.006	−0.001	0.006	0.008
**C**	0.003	−0.003	0.009	0.010
**D**	−0.003	−0.013	−0.003	0.014
RMSE	0.004	0.008	0.006	-

**Table 10 sensors-23-08199-t010:** The adjusted coordinates of **A,** assuming *ξ* = 0, *η* = 0.

Coordinates of A	
X (m)	3,835,763.324
Y (m)	1,177,324.807
Z (m)	4,941,576.311

## Data Availability

The data and the MathCad worksheet used in this study are available upon request from the co-author Edward Osada: edward.osada@dsw.edu.pl.

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
