# Peer review of "A Method for the Precise Coordinate Determination of an Inaccessible Location"

_sensors, 2023, doi:10.3390/s23198199_

Round 1

Reviewer 1 Report

The topic used a well-known spatial intersection method, which is analyzed with the use of out-of-plumb deflections, GNSS observations and two ways of rigorous adjustment. Each of the elements has a long timeline of research,  but the article uses them combined. It is an added value of the surveying element, but the main advantage is the sophisticated computation (adjustment).

Similar research was performed by: Skorupa B. (1997). Wyznaczenie współrzędnych punktów niedostępnych na podstawie opracowania pomiarów klasycznych i sygnałów GPS, z uwzględnieniem wpływu odchylenia pionu, Geodezja T. 3, UWND AGH, Kraków, p. 47–51. The author also analyzed the GNSS (GPS to be correctly) network, which was a base for total station measurements of inaccessible points (church towers). Two step adjustment was used (first the GPS network, then the angle network). Therefore, the adjustment was performed by less sophisticated then in the manuscript. Second work, with the use of spatial intersection, is the Borowski, L., Pienko, M. and Wielgos, P. (2017) ‘Evaluation of Inventory Surveying of Façade Scaffolding Conducted during ORKWIZ Project’, Proceedings - 2017 Baltic Geodetic Congress (Geomatics), BGC Geomatics 2017, pp. 189–192. Available at: https://doi.org/10.1109/BGC.Geomatics.2017.31. The work is rather simple examining of building scaffold, and comparing it to manufacture dimensions, but the authors have used similar equipment as in the manuscript (total station) and wrote a little about its precision. The manuscript does not describe this element widely. There is information about the standard deviation of coordinates (Table 2), which is not their accuracy. In the studies, the relative difference between points (1, 2, A) is important, not the absolute accuracy of each point on Earth. Nevertheless, a sort of pre-accuracy analysis (or just information), based on equipment class of accuracy, would benefit.   The third work is done by manuscript authors: Karsznia, Krzysztof, Edward Osada, and Zbigniew Muszyński. 2023. "Real-Time Adjustment and Spatial Data Integration Algorithms Combining Total Station and GNSS Surveys with an Earth Gravity Model" Applied Sciences 13, no. 16: 9380. https://doi.org/10.3390/app13169380. The topic is related, as the Materials and Methods has some similarities. It is suggest to referee to this works.

During the Polish Integration Campaign (Jaworski, L. (2011) Sprawozdanie techniczne IV etapu Kampani Integracyjnej - Opracowanie i wyrównanie obserwacji GNSS. Warszawa: CBK PAN.; Borowski, Ł. (2012) Determination of normal heights of ASG-EUPOS stations illustrated with an example of KRAW station, in: 4th Doctoral Seminar on Geodesy and Cartography, Biryło M. eds, Chapter: 2, Olsztyn: Wydawnictwo UWM.) the corresponding topic was analysed, but narrowed only to normal height determination.

There is no information about the thermal influence of the building on the refraction of light and, therefore, the deflection of the observed zenith angle. It was not the aim of the study. However, the authors are focused on precise coordinates determination, so this factor should be mentioned e.g. in the materials and methods chapter. Just a brief information about it is enough. If the measurements were done during the morning, it may be presumed that this factor was minimal.

The article has many equations that are hard to carefully trace, e.g. (s, alfa, beta, i, j) are shown in figure 1 (not so clear, but the task is very difficult to draw it in an easy way) and used later on. The table with all of the coefficients or observations is suggested. Later the “i” letter is used to describe (line 158) a set of coefficients (s, alfa,bea, X (…)). Suggestion: Change to another symbol e.g. “k”.

Line 82 – (…) e.g. EGM 2008 (…) - citation: Pavlis, N. K., Holmes, S. A., Kenyon, S. C., and Factor, J. K. (2012), The development and evaluation of the Earth Gravitational Model 2008 (EGM2008), J. Geophys. Res., 117, B04406, doi:10.1029/2011JB008916.

Figure 1  - the figure is based on: Edward Osada (2014), Geodezyjne pomiary szczegółowe. 2nd ed., Wrocław: UxLan, p. 398.

Chapter 2.3. point 3. The authors showed 2 ways to calculate out-of-plumb deflections and did not write which one was finally used in the experiment. Besides, why do astronomical latitude and longitude have to be computed, if the EGM2008's deflections may be calculated by Holmes and Pavlis 2006 program from spherical harmonic coefficients? see Pavlis, N. K., Holmes, S. A., Kenyon, S. C., and Factor, J. K. (2012), The development and evaluation of the Earth Gravitational Model 2008 (EGM2008), J. Geophys. Res., 117, B04406, doi:10.1029/2011JB008916.

The GNSS positioning is not a key factor here, the combining GNSS, total station observations, out-of-plumb deflections and sophisticated computations are. Therefore, it is suggested to change the title to e.g. “A Method for Precise Coordinates Determination of Inaccessible Location”. In the conclusions, the GNSS element is very limited, which is also an argument to change it.

Finally, minor or editing suggestions for the authors:

1)                  The analysis is difficult to read, a graph of the presented algorithm would be a very good explanation of the adjustment approach.

2)                  A table with a comparison of achieved differences would benefit. Now, each element is written separately and the tables consist of coordinates, not their differences. Therefore, the reader has to search what is the difference when the out-of-plumb deflections are used or when they are reduced to zero.

3)                  Please, use the same schema for all tables. One would like to check e.g. point A's GNSS RTK coordinates (Table 2) and compare them with GHM adjustment (Table 7), it may find it disturbed: the coordinates are in columns and then stitched to rows.

4)                  Reference items 1 and 6 are the same.

Reviewer 2 Report

Introduction

In line 36, you might want to specify what "observable" points are. Are these GNSS stations as well?

Line 38 could benefit from rephrasing for clarity: "The spatial intersection method, well-known to surveyors, allows for the determination of three-dimensional point coordinates based on angle and distance measurements from at least two points that can be occupied with total station/GNSS receivers (e.g., [1,2,3,4])."

In line 41, consider rephrasing for clarity: "examined the accuracy of determining coordinates of inaccessible points using, among other methods, the intersection method."

In lines 44-45, it might be helpful to provide a brief explanation or context about why the Gauss-Helmert model is relevant to the manuscript.

In line 46, consider specifying what "background information" is covered in textbooks [7] to provide more context.

Line 48 could benefit from specifying what properties of the Gauss-Helmert model have been analyzed in the literature.

In lines 49-51, mention what specific "related issues" have been studied by various authors.

In lines 52-54, briefly explain the significance or context of the advanced topics related to the Gauss-Helmert model.

Materials and methods

Line 66: It's mentioned that x coordinates are referenced to a local coordinate system, but it's not clear what these coordinates represent. Are they the coordinates of the total station or the target point? Please clarify.

Lines 70-74: While it's mentioned that the orientation angles of the local system are defined, it would be helpful to provide the actual definitions of these angles (Σ, ξ, η) in the text for clarity.

Lines 81-82: It's mentioned that geocentric coordinates and the Earth gravity model allow for the computation of deflection of the vertical. However, it's not clear how this computation is done or what equations/methods are used. Please provide more details or references.

Lines 90-96: The section discusses the least squares solution of intersection equations. It's unclear what these intersection equations represent or what they are used for. Please provide more context or background information.

Lines 113-115: The linearization of equations is discussed, but it's not clear how this linearization is done or what the Taylor series expansion involves. Please provide more details or references.

Lines 122-129: The Gauss-Helmert model is introduced, but it would be helpful to provide a bit more explanation or context about what this model is used for.

Lines 133-140: It's mentioned that the GNSS coordinates are used as controls, but it's not clear how these controls are incorporated into the calculations. Please provide more details.

Results

Line 237: It's good that you mention that GHM considers a priori accuracy, but it might be helpful to briefly explain why this is important or how it affects the results.

Lines 239-242: The process of using the LMA solution as approximate values for GHM could be explained in more detail. How were these values used to compute the matrix A, B, and the vector w?

Lines 247-250: While it's mentioned that GHM produced more accurate results compared to LMA, it would be beneficial to provide numerical values or percentages to quantify this improvement.

Lines 257-258: The statement about the "most significant ratio of corrections of adjusted observation v and their standard deviations is 2.8" could benefit from additional context or explanation.

Lines 261-264: It's mentioned that Experiment 2 was performed to verify the usability of GHM adjustment, but it would be helpful to briefly explain why this verification is important or what the objectives of Experiment 2 were.

Discussions

Lines 272-277: The introduction of the method and its connection to EGM 2008 is clear. However, it would be helpful to provide a brief overview of the key steps involved in this method before discussing its implications. This would give readers a clear roadmap of what to expect in the discussion.

Lines 278-282: The explanation of the case when ξ = 0 and η = 0 is concise and clear. However, it could be expanded slightly to explain why this simplification might be relevant in certain situations.

Lines 285-288: The discussion of the differences between the simplified (ξ = 0, η = 0) and normal solutions is informative. However, it would be helpful to provide a practical example or scenario where these differences might have a significant impact on the results. Additionally, what do you mean by "high vertical angles (> 40°)"? Please clarify the context of this statement.

Line 292: It's mentioned that the GHM provides a weighted LSQ solution and a rigorous accuracy assessment, which is an important point. However, it would be beneficial to explain briefly how this weighting is applied and why it matters in practical applications.

Some sentences are quite technical, and it might be beneficial to break them down or provide additional explanations.

Conclusions

While you mention that the proposed method allows for determining coordinates with an accuracy of approximately 1 cm in terms of standard deviation, it would be beneficial to briefly discuss the practical implications of this level of accuracy. For example, how does this level of accuracy compare to existing methods, and what types of applications or industries could benefit most from this precision?

You mention that the method can be utilized for engineering surveying and establishing ground control points for LiDAR and aerial photography. Expanding on this point by providing specific examples or case studies of real-world applications could help readers better understand the practical utility of your method.

It would be helpful to provide a brief comparison or discussion of how your method compares to other existing methods or techniques for determining coordinates of inaccessible locations. This can highlight the advantages and unique contributions of your approach.

It needs to be improved for better clarity and comprehensiveness.

Round 2

Reviewer 2 Report

Thanks for addressing all my comments. Try to improve the image's quality. Table 7. improve the distribution. Finally, avoid using "we, they" etc.